# Evaluating Impact of Emoticons and Pre-processing on Sentiment Classification of Translated African Tweets

**Saurav K. Aryal & Gaurav Adhikari**
Department of Computer Science
Howard University
Washington, DC 20059, USA
`saurav.aryal@howard.edu` & `gaurav.adhikari@bison.howard.edu`

## Abstract

This paper examines the impact of emoticons and pre-processing on sentiment classification for English translations of 11 African languages. Using AfriSenti-SemEval datasets, Roberta and Twitter-Roberta models are fine-tuned, and standard classification metrics are used to assess performance. The study concludes no significant performance differences with emoticons and pre-processing and no distinction between standard Roberta and domain-specific Twitter-Roberta.

## 1 Introduction

Africa has long faced developmental challenges due to climate and geopolitical issues(Gaillard & Mouton, 2022), resulting in a lack of funding and resources and a low output of research publications(**?**). Natural Language Processing (NLP) research has also been hindered by inadequate preservation and support for Africa's numerous languagesMartinus & Abbott (2019). While recent research has been performed for sentiment classification across multiple African languages Aryal et al. (2023), it has relied on models pre-trained on low-resource languages and did not evaluate the impact of emoticons and domain-specific preprocessing. This paper focuses on sentiment classification for 12 African languages by translating them into English and fine-tuning large English pre-trained models.

## 2 Methodology

We utilize the dataset for multilingual evaluation (ALL) from AfriSenti-SemEval(Muhammad et al., 2022; Yimam et al., 2020), which comprises tweets in Hausa(HA), Yoruba(YO), Igbo(IG), Nigerian Pidgin (PCM), Amharic(AM), Algerian Arabic(DZ), Moroccan Arabic/Darija(MA), Swahili(SW), Kinyarwanda(KR), Twi(TWI), Mozambican Portuguese(PT), Xitsonga(Mozambique Dialect) (TS). The dataset was translated into English using Google Translate API, and three experiments were performed. The experiments removed emoticons, @*User* tags, or retained both while standardizing other data preprocessing, such as removing HTML tags, punctuations, URLs, white spaces, non-alphabetic characters, accented characters, and digits. Each model was independently trained on these experimental conditions. These experiments evaluate the effect of removing emoticons and domain-specific preprocessing on the model's performance. With default hyperparameters, two variants of the Bidirectional Encoder Representations from the Transformers (BERT) model: Roberta (Robustly Optimized BERT Approach)Face (b), which includes a larger batch size, training times, and training data than BERT and domain-specific Twitter-Roberta Face (a), which is pre-trained on Twitter data, including a significant amount of emoticons and domain-specific text. Experiments and models report performance with standard macro-averaged classification metrics: accuracy, precision, recall, and F1-score. We also compare approaches using Precision-Recall (PR) curves to evaluate models and class-wise performance.

## 3 RESULTS

No significant differences are found in the performance of our model choices, as seen in Table 1. Surprisingly, no performance difference was noticed between the experiments with and without emoticons and user tags. The PR curve in Appendix 2 further supports this observation. Thus, the impact of emoticons and *@user* tags may not be significant for sentiment classification for English translations of African Languages. Next, we examine the class-level PR curve for the best-performing model in Figure 1. The plot and attached Average Precision (AP) scores demonstrate that the model is significantly better at discriminating positive sentiments from either positive or neutral. It may be possible that translations from Google API are better or more accurate for positive sentiments. For further analysis, we also report the performance on the test set separated by language and experiment in Tables 2, 3, 4. Observations suggest that languages with larger sample sizes usually perform better than those with smaller ones. We also share our trained models and source in the Appendix.

Table 1: Performance Metrics for Train and Test Splits

| Experiments | Models | Train | | | | Test | | | |
|---|---|---|---|---|---|---|---|---|---|
| | | Recall | Precision | Accuracy | F1 | Recall | Precision | Accuracy | F1 |
| emoji + user tag | Roberta | 0.66 | 0.66 | 0.66 | 0.65 | 0.65 | 0.65 | 0.65 | 0.65 |
| | T-Roberta | 0.67 | 0.68 | 0.67 | 0.67 | 0.66 | 0.67 | 0.66 | 0.66 |
| emoji removed | Roberta | 0.66 | 0.66 | 0.66 | 0.66 | 0.66 | 0.66 | 0.66 | 0.66 |
| | T-Roberta | 0.66 | 0.68 | 0.66 | 0.66 | 0.66 | 0.67 | 0.66 | 0.65 |
| user tag removed | Roberta | 0.65 | 0.66 | 0.65 | 0.65 | 0.65 | 0.66 | 0.65 | 0.65 |
| | T-Roberta | 0.67 | 0.67 | 0.67 | 0.66 | 0.66 | 0.67 | 0.66 | 0.66 |

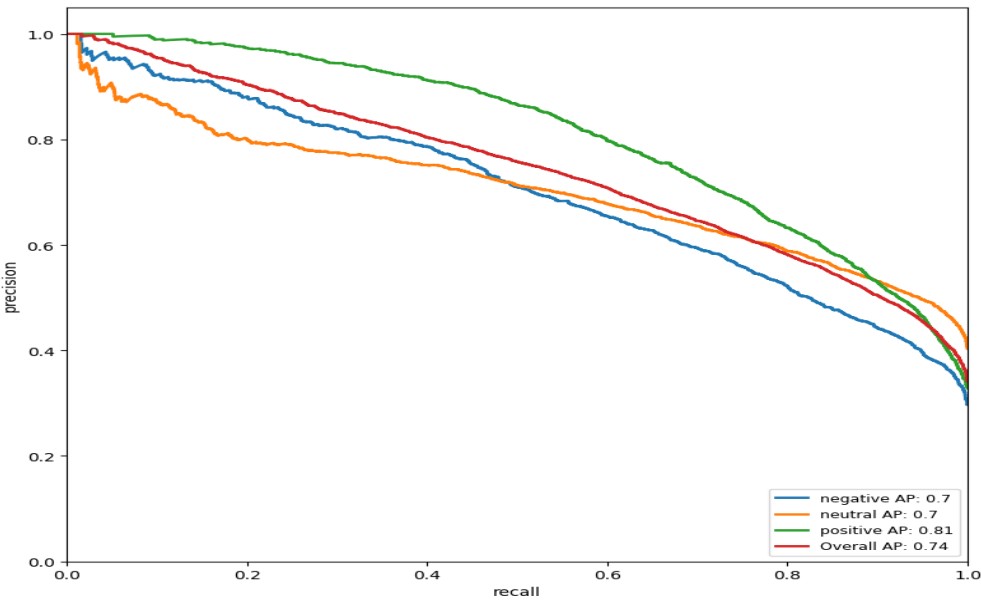

Figure 1: Precision-Recall Curve of Best-performing model for each class

## 4 CONCLUSION

Surprisingly, no significant performance difference was noticed between the experiments with and without emoticons and user tags. These results suggest their impact is insignificant for sentiment classification using English translations of African languages. The best-performing model seemed better at discriminating positive sentiments than negative or neutral, indicating that translations from Google API may be more accurate for positive sentiments. However, more research must build on

current work and improve predictive performance for sentiment analysis in general. Current limitations and future work include evaluating word-level translation for handling potential transliterations and code-switching, hyperparameter optimization, and datasets unbalanced by languages.

URM STATEMENT

The authors acknowledge that all key authors of this work meet the URM criteria of ICLR 2023 Tiny Papers Track.

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

## A APPENDIX

Please feel free to use our public google drive folder with our trained models and source code to reproduce our work HERE!. Please do not hesitate to contact the authors with any questions or feedback.

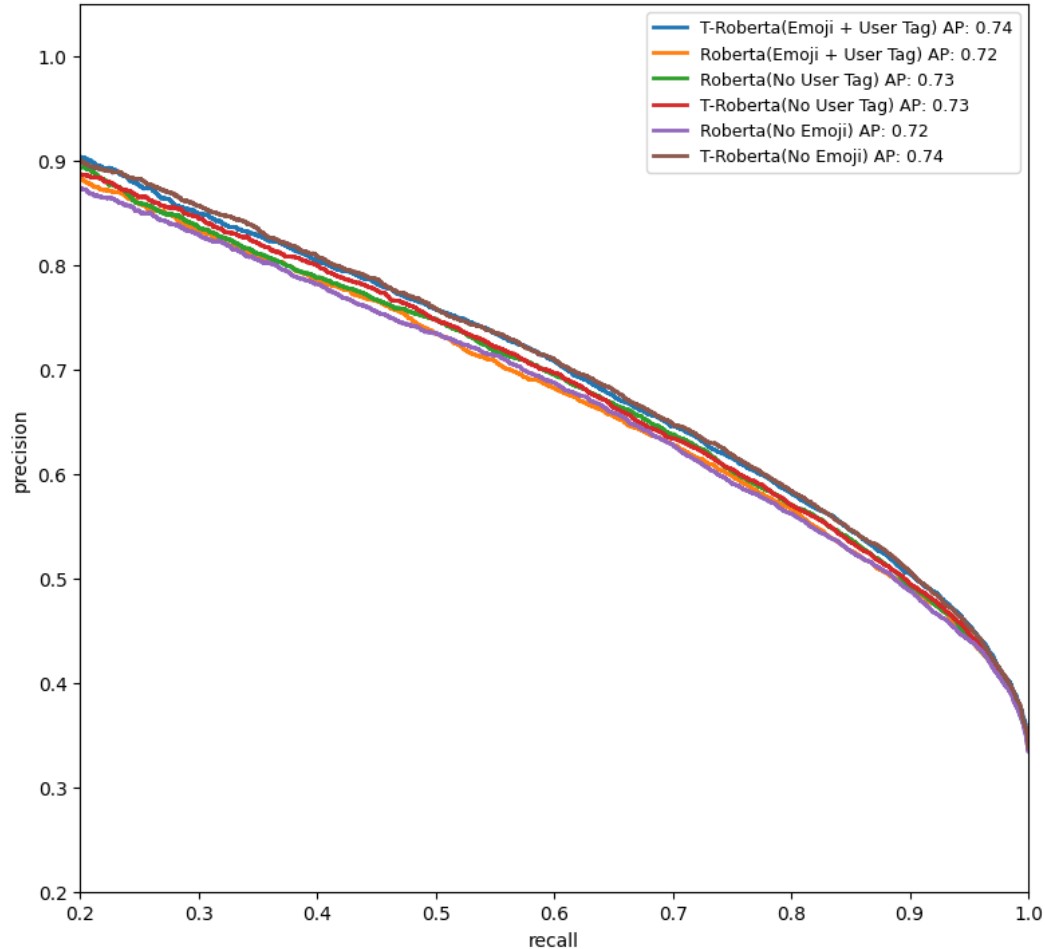

Figure 2: Precision-Recall Curve comparing all Models and Experiments

Table 2: Test Results for Each language: Models Trained with Emoji  User Tag Retained

| | | | | Test | | | | |
|---|---|---|---|---|---|---|---|---|
| | **Roberta** | | | | **T-Roberta** | | | |
| | **Accuracy** | **Recall** | **Precision** | **F1** | **Accuracy** | **Recall** | **Precision** | **F1** |
| **KR** | 0.62 | 0.62 | 0.65 | 0.62 | 0.63 | 0.63 | 0.66 | 0.62 |
| **TWI** | 0.50 | 0.50 | 0.55 | 0.50 | 0.49 | 0.49 | 0.56 | 0.50 |
| **SW** | 0.61 | 0.61 | 0.61 | 0.61 | 0.57 | 0.57 | 0.58 | 0.58 |
| **IG** | 0.65 | 0.65 | 0.66 | 0.64 | 0.67 | 0.67 | 0.67 | 0.66 |
| **DZ** | 0.52 | 0.52 | 0.61 | 0.53 | 0.53 | 0.53 | 0.66 | 0.55 |
| **HA** | 0.72 | 0.72 | 0.73 | 0.72 | 0.74 | 0.74 | 0.75 | 0.74 |
| **YO** | 0.71 | 0.71 | 0.71 | 0.71 | 0.72 | 0.72 | 0.73 | 0.71 |
| **MA** | 0.66 | 0.66 | 0.69 | 0.65 | 0.67 | 0.67 | 0.69 | 0.66 |
| **PT** | 0.61 | 0.61 | 0.65 | 0.62 | 0.67 | 0.67 | 0.68 | 0.67 |
| **AM** | 0.58 | 0.58 | 0.59 | 0.58 | 0.61 | 0.61 | 0.61 | 0.61 |
| **TS** | 0.48 | 0.48 | 0.55 | 0.49 | 0.46 | 0.46 | 0.56 | 0.47 |

Table 3: Test Results for Each language: Models Trained with Emoji Removed

| | Test | | | | | | | |
| | Roberta | | | | T-Roberta | | | |
| | Accuracy | Recall | Precision | F1 | Accuracy | Recall | Precision | F1 |
|---|---|---|---|---|---|---|---|---|
| **KR** | 0.62 | 0.62 | 0.68 | 0.62 | 0.62 | 0.62 | 0.68 | 0.62 |
| **TWI** | 0.47 | 0.47 | 0.57 | 0.50 | 0.47 | 0.47 | 0.57 | 0.50 |
| **SW** | 0.59 | 0.59 | 0.58 | 0.58 | 0.59 | 0.59 | 0.58 | 0.58 |
| **IG** | 0.68 | 0.68 | 0.69 | 0.68 | 0.68 | 0.68 | 0.69 | 0.68 |
| **DZ** | 0.49 | 0.49 | 0.64 | 0.50 | 0.49 | 0.49 | 0.64 | 0.50 |
| **HA** | 0.73 | 0.73 | 0.74 | 0.73 | 0.73 | 0.73 | 0.74 | 0.73 |
| **YO** | 0.72 | 0.72 | 0.74 | 0.72 | 0.72 | 0.72 | 0.74 | 0.72 |
| **MA** | 0.67 | 0.67 | 0.69 | 0.66 | 0.67 | 0.67 | 0.69 | 0.66 |
| **PT** | 0.65 | 0.65 | 0.66 | 0.66 | 0.65 | 0.65 | 0.66 | 0.66 |
| **AM** | 0.61 | 0.61 | 0.61 | 0.61 | 0.61 | 0.61 | 0.61 | 0.61 |
| **TS** | 0.44 | 0.44 | 0.57 | 0.47 | 0.44 | 0.44 | 0.57 | 0.47 |

Table 4: Test Results for Each language: Models Trained with User Tag Removed

| | Test | | | | | | | |
| | Roberta | | | | T-Roberta | | | |
| | Accuracy | Recall | Precision | F1 | Accuracy | Recall | Precision | F1 |
|---|---|---|---|---|---|---|---|---|
| **KR** | 0.62 | 0.62 | 0.65 | 0.62 | 0.62 | 0.62 | 0.67 | 0.61 |
| **TWI** | 0.44 | 0.44 | 0.53 | 0.46 | 0.35 | 0.35 | 0.53 | 0.38 |
| **SW** | 0.61 | 0.61 | 0.61 | 0.61 | 0.59 | 0.59 | 0.58 | 0.58 |
| **IG** | 0.67 | 0.67 | 0.67 | 0.66 | 0.67 | 0.67 | 0.68 | 0.67 |
| **DZ** | 0.50 | 0.50 | 0.62 | 0.52 | 0.50 | 0.50 | 0.65 | 0.51 |
| **HA** | 0.72 | 0.72 | 0.72 | 0.72 | 0.74 | 0.74 | 0.75 | 0.74 |
| **YO** | 0.72 | 0.72 | 0.73 | 0.72 | 0.70 | 0.70 | 0.74 | 0.69 |
| **MA** | 0.66 | 0.66 | 0.69 | 0.65 | 0.66 | 0.66 | 0.72 | 0.65 |
| **PT** | 0.66 | 0.66 | 0.68 | 0.66 | 0.66 | 0.66 | 0.67 | 0.66 |
| **AM** | 0.60 | 0.60 | 0.61 | 0.60 | 0.62 | 0.62 | 0.62 | 0.62 |
| **TS** | 0.43 | 0.43 | 0.56 | 0.46 | 0.44 | 0.44 | 0.62 | 0.47 |

