# OpenReview forum: "Evaluating Impact of Emoticons and Pre-processing on Sentiment Classification of Translated African Tweets"
_ICLR.cc/2023/TinyPapers — Submitted to Tiny Papers @ ICLR 2023_

### Official Review · Reviewer_zWNv · 2023-03-21

**Confidence:** 4

**Summary Of Contributions:**

This paper evaluates the impact of emoticons as well as other artifacts typically removed in preprocessing such as punctuation, user tags, html tags, URLs, and more. After translating 11 different African languages to English and fine tuning state of the art models, they found that the presense and absense of the emoticons and user tags had no implication on the performance of the models.

**Rating:**

High Potential (HP): a submission which meets the reviewing criteria and has potential to make an impact on the field

**Strengths And Weaknesses:**

Strengths

- Clarity: Results and methodology are communicated clearly and effectively.

- Correctness: Methodology is sound and evaluates the hypothesis well. Multiple models are evaluated and a variety of metrics beyond accuracy are used to ensure a clear and complete interpretation of the results.

- Reproducibility: A nice and simple ablation study is performed that would be easy to replicate. Details are provided on datasets used, models trained, and preprocessing steps taken. Source code is also provided.

- Follows basic requirements: Is within the page limit and follows the code of conduct


Weaknesses

- Clarity/Correctness: The methodology is well laid out and evaluates the proposed hypothesis well. However I think the choice to translate the samples to English could be better justified. Additionally, it would be great to have a few more details on the emoticons and how they were used.


**Suggested Changes:**

Thank you for your submission! I enjoyed reading your paper and thought it was an interesting question to answer. Overall, great job! I just have a few suggested changes that I think could improve the quality of the paper:

As stated in the weaknesses, I think a simple (even just a sentence) addition to the paper that justifies why the samples were translated to English first would improve the clarity of this portion of the experiment. I feel like there may be nuances of the languages that get lost in the translation process, so this choice was unintuitive to me at first glance.

I was also curious what kinds of emoticons were present in the data; were these complete emojis or simple text-based ones such as :) ? In either case, was there any preprocessing done on these before feeding them into the model? I think adding these details would also enhance the clarity of the methodology and allow for a more complete understanding of the experiment.


Minor changes/Typos:

- The abstract states that 11 African languages were evaluated but the intro states 12 and lists out 12 languages.

- The first sentence of the intro seems to be missing a cite or has a broken link, as there is a (?) at the end.

---

### Official Review · Reviewer_zfWY · 2023-04-02

**Confidence:** 4

**Summary Of Contributions:**

 The paper presents a study to evaluate the effect of removing emoticons and domain-specific preprocessing on the English Translated African tweets for sentiment Classification for 11 African Languages

**Rating:**

Clear, Correct, and Reproducible (CCR): a submission which meets the reviewing criteria

**Strengths And Weaknesses:**

Strengths
1. The motivation is clear and correct
2. The experimental design is intuitive and the methodology is easy to understand and reproduce (simple study with all details and source code provided)
3. The paper evaluates multiple models to prove the hypothesis and use reasonable metrics for their analysis


Weakness
1. Despite a clear and correct motivation built throughout the paper, it lacks a solid background for translating tweets to English and further study in the Translation model's performance for the different languages



**Suggested Changes:**

1. I would like to see the language specific translation performance of the Google Translate API, I believe the performance would be different for different languages and should be significant to further understand the sentiment classification results for these African languages

2. The experimental design is clear and correct however, adding the intuition behind translating to English would make the experimental design stronger

3. I would also like to see some details or examples about the kind of emoticons used in the tweets and additional details about the preprocessing techniques

## Typos & Minor Edits
1. 11 languages in abstract when 12 languages listed in Section 2
2. Missing citation (?) in Section 1
3. The x and y labels of Figure 1 are not very clear to read

---

### Meta-Review · Area_Chair_56yK · 2023-04-06

**Recommendation:** Invite to present
**Confidence:** 4

**Metareview:**

 The paper explores ways to evaluate the effect of removing emoticons and domain-specific preprocessing on African tweets (translated into English) for sentiment Classification for 12 African Languages.
The paper is well-presented. The hypothesis is supported by nice set of experiments and comparison with a few baselines. The contribution of the paper is valuable. A more in-depth analysis and evaluation is missing: a solid background for translating tweets to English and further study in the translation model's performance for the different languages.  Additionally, it would be great to have a few more details on the emoticons and how they were used.


**Summary:**

Translate and remove/retain emoticons on Tweets using different Roberta models. More in-depth analysis and evaluation is missing.

**Comments And Feedback To The Authors:**

Cool idea, nicely written and presented. By addressing reviewers' comments and questions, this paper can become a good contribution to the literature.



**Reason For Not Giving A Higher Recommendation:**

Despite a clear and correct motivation built throughout the paper, it lacks a solid background for evaluation of the translation aspect.


**Reason For Not Giving A Lower Recommendation:**

The paper is well-presented, the hypothesis is well-supported, and the question is an interesting one for the field.

---

### Decision · Program_Chairs · 2023-04-07

Invite to present